# DISCRIMINATION FOR GENERATION: GENERAL SCHEMA FOR UTILIZING DISCRIMINATIVE MODELS FOR GENERATIVE MODELS

## ABSTRACT

There are two primary approaches to learning from data: discriminative models, which make predictions based on provided data, and generative models, which learn data distributions to create new instances. This paper introduces a novel framework, Discrimination for Generation (DFG), as the first attempt to bridge the gap between discriminative and generative models. Through DFG, discriminative models can function as generative models. We leverage Neural Tangent Kernel (NTK) to map discriminative models into a connected functional space, enabling the calculation of the distance between the data manifold and a sampled data point. Our experimental results demonstrate that the proposed algorithm can generate high-fidelity images and can be applied to various tasks such as Targeted Editing and Inpainting, in addition to both unconditional and conditional image generation. This connection provides a novel perspective for interpreting models. Moreover, our method is algorithm-, architecture-, and dataset-agnostic, offering flexibility and proving to be a robust technique across a wide range of scenarios.

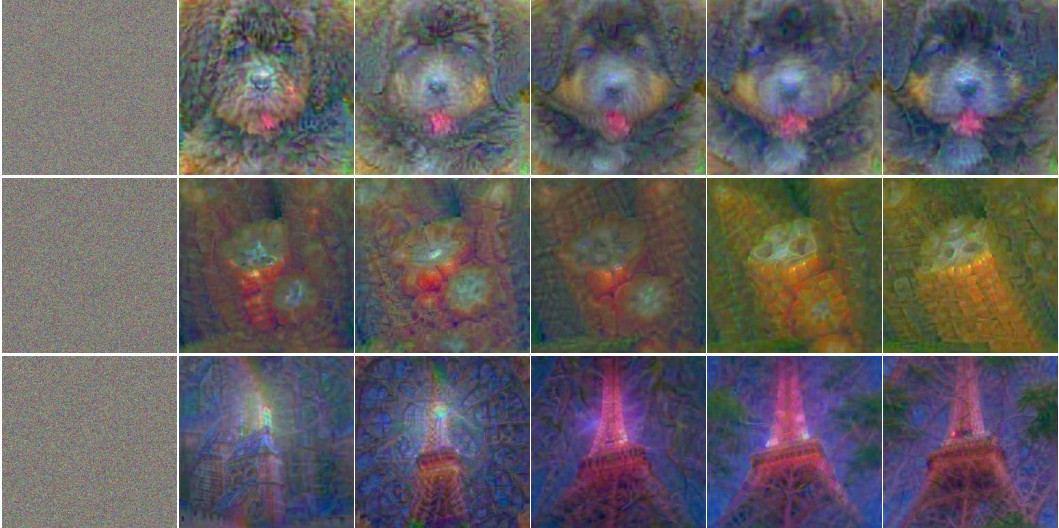

Figure 1: From left to right, the image shows the generation process described in Eq. (**??**) using pretrained DINOv2, a discriminative model, without any condition. No generative models like diffusion models were used; instead, we only leverage pretrained discriminative models to generate images.

## 1 INTRODUCTION

There are mainly two approaches to learning from data. Discriminative approaches use sampled train set to predict properties of test set from unknown distribution, and generative approaches generate new instances based on the distribution learned from sampled data. The two approaches are related

in the sense that they are trained to understand the true distribution of data. Discriminative models aim to make accurate predictions even for unseen data by learning patterns from the sample data, which it generalizes under the assumption that it represents the entire distribution. Generative models strive to replicate or create new data points that reflect the broader characteristics of the actual data distribution, going beyond the limited set of observed samples. In other words, these methodologies are equivalent because their ideal goal is to understand the 'true' data distribution. By empirically understanding the connection between the two, we can obtain an important insight regarding how deep learning handles data. Nevertheless, current line of researches draws light to only one direction of the equivalence. There have been works that directly utilize generative models as discriminative models, (Li et al., 2023; Kingma et al., 2014a), while the other way around – using discriminative models as generative ones – is rarely discussed (Haim et al., 2022; Lee et al., 2024).

There have been successful works that demonstrated how generative models such as VAE (Kingma & Welling, 2022) and diffusion models (Ho et al., 2020a) can be used as discriminators (Kingma et al., 2014b; Li et al., 2023). Despite the successful conversion from a generative model to a discriminative model, attempts to conduct generation with discriminative models are scarce. Similar works focus solely on training data reconstruction through model inversion (Wang & Kurz, 2022), which cannot generate unseen images within the data manifold. The work on Deep Support Vectors (DSVs) (Lee et al., 2024) is an attempt to find support vectors in any deep neural network. DSVs effectively generates visually plausible images even from high-dimensional data, and since the generated images visualize hidden decision criteria rather than being mere reconstructions, DSVs can be considered a generative approach. However, DSVs can only generate images conditioned on class labels, which limits its classification models, making it difficult to classify it as a true generative model. Additionally, it lacks a mathematically plausible explanation. These limitations are also present in related works. On the other hand, our algorithm **overcomes this issue**.

Our work ambitiously pioneers a way to transform discriminative models into generative models. We leverage the Loss Tangent Kernel (LTK) (Chen et al., 2023), a variant of Neural Tangent Kernel (NTK) (Jacot et al., 2018) to map the tuple of a dataset, training algorithm, and neural network architecture to a trained network. This enables us to generalize the score function of diffusion models, allowing it to compute loss directly from the trained model. The entire process is grounded in firm mathematical justification. To the best of our knowledge, our work is the first attempt to integrate NTK and the score function for utilizing discriminative models in generative tasks. Our method goes a step further in achieving practical efficiency. While generative models are notoriously challenging to train (Salimans et al., 2016) our approach offers the advantage of directly applying already-trained discriminative models for generation without additional training. Although the performance may not yet surpass state-of-the-art generative models, our method demonstrates that nearly all types of discriminative models can function as generative models, as shown through extensive experiments. In addition, we provide an eXplainable AI (XAI) method by visualizing specific features using only model parameters. This innovative approach represents a shift from traditional feature visualization techniques in model explainability.

This paper presents Discrimination for Generation(DFG): a method for utilizing discriminative models in generation tasks and explains the approach in a mathematical framework. The mathematical explainability is key advantage of our method over visualization of DSV (Lee et al., 2024). We show that any discriminative model can be transformed into a generative model using the universal methodology. The generalization of DFG is validated thoroughly through extensive experiments.

We generalize the score-matching problem from probability space to the one involving arbitrary measures. The problem of calculating the distance of a data point to a manifold in the input space is transformed into the problem of calculating the distance in the functional space. We then show that this process can be written in the same form as the objective function of a conditional diffusion model. This is supported by experiments using various architectures, including ViT (Dosovitskiy et al., 2021), ResNet (He et al., 2015), and DETR (Carion et al., 2020). Different training methodologies – such as self-supervised learning, classification, and object detection – were applied to demonstrate that the proposed approach can be applied universally. In the end, we show that we can generate images from discriminative models similarly to diffusion models.

Our contributions are as follows:

1. We provide a methodology that can transform a common discriminative model into a generative model without any post-training process.

2. We show that our method works in a similar manner to a diffusion model, and is universally applicable to any architecture, algorithm and dataset.

## 2 RELATED WORKS

**Score-Based Generative models** generates by approximating the actual probability of the data it wants to generate (Hyvärinen & Dayan, 2005; Song et al., 2020b). The approximation cannot be made directly, so diffusion process is employed. The process adds noise to target distribution to create a distribution we know of – for example, Gaussian distribution (Ho et al., 2020b; Song et al., 2020a) – and reverse the process to obtain a solution. In a nutshell, score-based generative models generate images by iteratively moving from Gaussian distribution towards image manifold. Training models to solve this problem requires delicate methodologies; learning a score function instead of function distribution (Ho et al., 2020a), handling data differently at each time-step. Noisy classification is also necessary to perform different classifications at different time steps.

Our method Discrimination for Generation(DFG) fundamentally solves the probability approximation task the score-based model wanted to solve but failed due to complexity. We directly approximate the distance between data manifold and a sampled image by leveraging Neural Tangent Kernel (Jacot et al., 2018). Based on empirical and theoretical evidence that the discriminative model inherently understands the data manifold, we leverage this to compute a generalized distance. This way our methodology allows the use of a pretrained discriminative model as a plug-and-play generative model without the need for additional training or new architectures. Similarly, the conditioning process does not require any noisy guidance.

**Model Visualization** diverges into two lines of research. First, gradient-based method is used in Feature Visualization, one of the approaches in eXplainable AI (XAI) (Zeiler & Fergus, 2014), where images are generated to maximize a model's specific output, such as neurons and layers (Erhan et al., 2009) to understand how the model makes decisions (Mahendran & Vedaldi, 2015; Fel et al., 2023). A fundamental problem with this method is that it relies on gradient ascent. We cannot claim that such visualizations accurately reflect the model's decision criteria. One example is adversarial attack, which also exploits gradient ascent. The visualizations here manipulate inputs to maximize specific objectives, however, the results do not provide any insights on decision criteria. XAI methods may generate plausible images, yet it is still uncertain whether these images truly reflect the model's decision boundary.

Second is maximal-margin based technique. It employs stationarity condition along with the equivalence between Support Vector Machine(SVM) and deep-learning models to reconstruct training dataset (Lyu & Li, 2020). Since it is an application of SVM, most works are conducted in SVM-like settings. Their work is usually limited to binary classification (Haim et al., 2022) and uses only small fraction of dataset (Buzaglo et al., 2023). The setting is also extremely overfitted and restricted to small dimensional data such as CIFAR10 and MNIST with simple architecture like ConvNet and MLP (Goodfellow et al., 2016). Although Deep Support Vector (Lee et al., 2024) is relatively free from such limitation discussed above, it is still bounded to classification tasks since it is based on SVM. Naturally, it can generate images only when class labels are given as condition, requiring a label pair for image generation. This consequence implies DSV lacks 'true' generation ability as it cannot estimate unconditional image distribution.

Discrimination for Generation overcomes the limitations the two model visualization methods have. Unlike conventional gradient-based methods that focus on maximization of specific neurons or layers, DFG leverages the entire model and replaces gradient ascent with functional approximation. This ensures that the generated data optimally approximates the existing manifold, making it the objective to provide the model's decision criteria based on realistic images aligned with the underlying manifold. These characteristics provide justification to apply our visualization on XAI area. Also, DFG is free from all restrictions maximal-margin based technique possesses. We now insist we have successfully transformed various discriminative architectures such as those for classification, SSL (Self-Supervised Learning) and object detection to generative models in an architecture-, algorithm-, dataset-agnostic way.

## 3 PRELIMINARIES

**Kernel Methods** are conventional machine learning techniques where a bivariate kernel $k(\cdot, \cdot)$ maps two data points in a high-dimensional feature space to inner product $k(x, y) = \langle \phi(x), \phi(y) \rangle$. Reproducing Kernel Hilbert Space (RKHS) guarantees the existence of the space $\phi$ where the closeness of $\phi(x)$ and $\phi(y)$ implies the closeness of $x$ and $y$. Fixing one of the inputs, we get a univariate kernel $k(\cdot, x)$. It is worth noting that this kernel exists in a functional space, i.e., it is a function.

**Neural Tangent Kernel** tries to explain the generalization ability of deep learning models using conventional kernel methods. It suggests that for every deep learning network being trained, a corresponding kernel exists. The kernel is defined by the gradients of the given deep learning model, $\Theta(x, x') = \nabla_\theta f(x)^\top \nabla_\theta f(x')$, reflecting how the model trains in the input space. However, as NTK does not account for loss dynamics, Loss Tangent Kernel Chen et al. (2023) extends NTK by incorporating the derivative of the loss function. The LTK is expressed as: $K(x, x') = \nabla_f \ell(f(x; \theta))^\top \Theta \nabla_f \ell(f(x'; \theta))$.

## 4 DISCRIMINATION FOR GENERATION

In this section, we introduce our algorithm Discrimination for Generation (DFG). The introduction is done step by step. First, we generalize the computation method of the score model in Sec. 4.1 and argue that the goal of this framework is to minimize the distance between the generated image and the data manifold. We revisit the idea that a discriminative model actually perceives the manifold when it achieves generalization ability in Sec. 4.2. Then in Sec. 4.3 we introduce the concept of Functor to which we can apply distance measure. We demonstrate that the Neural Tangent Kernel (NTK) is a suitable Functor for the purpose. Finally, we explain DFG in Sec. 4.4, the metric that calculates distance between images and the manifold, using what have been discussed throughout the three previous steps.

### 4.1 SCORE-BASED GENERATIVE MODEL

Designing a generative model includes estimating data distribution $p(x)$ using a model $f$. There are two important points to consider. One is that $p(x)$ is unknown. What we do know is the sampled dataset $\mathcal{D}_s \sim p(x)$. Another is, the image space $\mathbb{R}^{hw}$ is so high-dimensional that the manifold $\mathcal{M} \subseteq \mathbb{R}^{hw}$ where actual data resides in is relatively small and hard to approximate.

In a sampling process of score-based generative model, we first sample $x_0$ from a known distribution and iteratively update it for $T$ steps to generate $x_T$. The resulting output is expected to be placed within the manifold $x_T \in \mathcal{M}$. The update from $x_0$ to $x_T$ is restricted so that the sample approaches the manifold $\mathcal{M}$ every step. In other words, $d(x_{t+1}, \mathcal{M}) < d(x_t, \mathcal{M})$ must hold for some distance metric $d$. Finding the proper $d$ is critical to successfully lure the sample towards the data manifold. Sampling process can be understood as an optimization process that reduces the distance in this sense.

Since we have no access to $\mathcal{M}$, the above process should be done using an approximation using $\mathcal{D}_s \subseteq \mathcal{M}$. This can be written as the following formula.

$$x_{t+1} = x_t - \nabla_x d(x, \mathcal{M}) := x_t - \nabla_x \int_\mathcal{M} d(x_t, x) dx \simeq x_t - \nabla_x \sum_i d(x_t, \zeta_i), \quad (1)$$

where, $\zeta_i$ is the $i$th sample of $\mathcal{D}_s$.

Direct implementation of the equation is almost impossible. There are two reasons for this: 1. It is difficult to find a metric $d$ that can well-represent the manifold, $\mathcal{M}$, since the manifold takes up only a small portion in the high-dimensional space and its shape is very curvy. 2. The amount of calculation is proportional to the number of samples in our data $\mathcal{D}_s$, *i.e.*, proportional to $|\mathcal{D}_s|$.

Score-based models resolve the difficulties by translating Eq. (1) from the geometric framework $\mathcal{M}$ to a probability space. This shifts the perspective from a geometric view to a probabilistic measure. The resulting equation, now framed within the probability space, is provided below:

$$x_{t+1} = x_t + \nabla_x \log p(x_t). \quad (2)$$

Here, the term score function refers to the gradient of the log probability density function, $\nabla_x \log p(x_t)$, which can be replaced by $-\nabla_x d(x, \mathcal{M})$. The score-based model approximates this Jacobian, $\nabla_x \log p(x_t)$, using a deep learning model $s_\theta(x_t, t)$ (Ho et al., 2020a).

## 4.2 DISCRIMINATIVE MODELS ALREADY HAVE KNOWLEDGE ABOUT MANIFOLD

If we aim to compute Eq. (1) directly, the question we must address is straightforward: Can we generalize it? With only limited data available, we need an ideal distance function, $d$, that meets the following condition to ensure generalization.

$$d(\cdot, \mathcal{D}_s) \simeq d(\cdot, \mathcal{D}_t), \quad \forall \mathcal{D}_t \subseteq \mathcal{M}, \tag{3}$$

where $\mathcal{D}_s$ is a train set and $\mathcal{D}_t$ is a test set.

We already know a number of models that **have to** satisfy the above condition: the discriminative models. Discriminative models, though only trained with train set, are required to demonstrate similar accuracy on test set $\mathcal{D}_t$ as on train set $\mathcal{D}_s$. This is the very property we wish for $d$ to have. Now the next question is, **how can we transfer this property to the distance function $d$?**

## 4.3 INTRODUCING THE FUNCTOR $\mathcal{F}$

The generalization ability of a deep learning model through optimization does not depend on the specially-designed train set; rather, it only requires that the set $\mathcal{D}_s$ exists within $\mathcal{M}$ and is an iid sample. In other words, as long as a few basic conditions are satisfied (Goodfellow et al., 2016), the model will generalize well to $\mathcal{D}_t$. From this, we can derive the following intuition: taking two deep neural networks, $f_1$ and $f_2$, and training each with different dataset, $D_1$ and $D_2$, the resulting models **will act similarly** if $D_1$ and $D_2$ are in the same manifold. Now, imagine a transformation that maps a train data to a trained model. The transformation would be a mapping that sends **similar** data, i.e. data in the same manifold, to **similar** functions and enable the use of distance metric in the functional space as the proxy for distance metric in the image space.

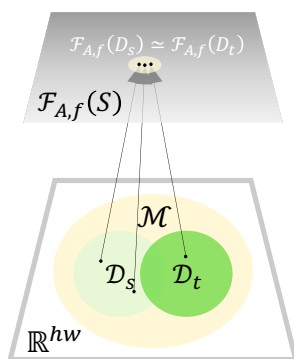

Figure 2: Structure of Functor $\mathcal{F}_{\mathcal{A},f}$. $\mathcal{D}_s$ and $\mathcal{D}_t$ sampled from the manifold is mapped from $\mathcal{R}^{hw}$ space to $\mathcal{F}_{\mathcal{A},f}(S)$, $\forall S \subseteq \mathbb{R}^{hw}$ space. The two are mapped to similar space due to the generalization property of discriminative model.

To leverage this abstract property, we introduce the concept of Functor (MacLane, 1998) – essentially a transformation from one structure to another – that enables us to exploit the generalization characteristic effectively.

Let us define a Functor $\mathcal{F}$ that takes as input a tuple of a train dataset $\mathcal{D}_s$, a neural network $f$, and an algorithm $\mathcal{A}$ such as SGD that would train $f$ and returns as output a trained neural network $f^\star$.

$$\mathcal{F}: \quad (\mathcal{D} \times \mathcal{A} \times f) \to f^\star \tag{4}$$

Fixing $f$ and $\mathcal{A}$, to ResNet-50 and SGD for example, we obtain a Functor that maps data to a function, $\mathcal{F}_{\mathcal{A},f}: \mathcal{D} \to f^\star$. Now we can write the desiderata for a good model that generalizes well and ensures similar evaluation for both train set $D_s$ and test set $D_t$ (see Fig. 2):

$$\mathcal{F}_{\mathcal{A},f}(\mathcal{D}_s) \simeq \mathcal{F}_{\mathcal{A},f}(\mathcal{D}_t) \quad \forall \mathcal{D}_s, \mathcal{D}_t \subseteq \mathcal{M}. \tag{5}$$

Assume that the minimization of distance between Functors also minimizes the distance between Functor inputs. Under the assumption we can replace the distance metric, $d(x, \mathcal{M})$, in Eq. (1) with that of the Functor, $d(\mathcal{F}_{\mathcal{A},f}(x), \mathcal{F}_{\mathcal{A},f}(\mathcal{M}))$. If a trained model has learned to generalize and works well on any test set, $\mathcal{F}_{\mathcal{A},f}(\mathcal{M})$ here can be replaced with $\mathcal{F}_{\mathcal{A},f}(\mathcal{D}_s)$.

To sum up, Eq. (1) can be written in the following way:

$$x_{t+1} = x_t - \nabla_x d(\mathcal{F}_{\mathcal{A},f}(x), \mathcal{F}_{\mathcal{A},f}(\mathcal{M})) \simeq x_t - \nabla_x d(\mathcal{F}_{\mathcal{A},f}(x), \mathcal{F}_{\mathcal{A},f}(\mathcal{D}_S)). \tag{6}$$

We will now use the Functor $\mathcal{F}$ for optimization. Optimization would reduce the distance in the functional space. For our previous assumption to be true so that the minimization ensures the reduction of distance between inputs, $\mathcal{F}_{\mathcal{A},f}$ is required to be continuous in topological space. Indeed, we

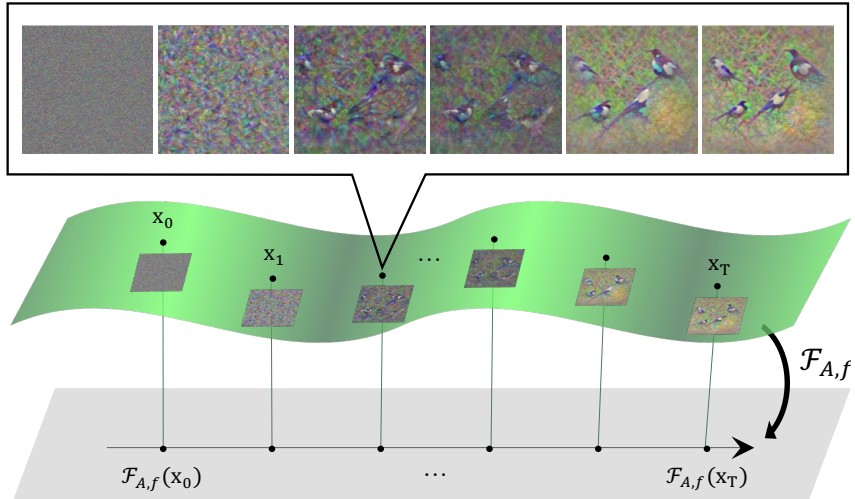

Figure 3: Conceptual diagram of DFG process. Similar to the mirror gradient descent, we update image $x_t$ using the distance metric in the functional space defined by the Functor $\mathcal{F}_{\mathcal{A},f}$.

can treat the space as continuous. Numerous studies (Bousquet & Elisseeff, 2002; Elisseeff et al., 2005; Xu & Mannor, 2012; Hardt et al., 2016) support the fact that when small changes occur in the dataset, the models trained on them also exhibit only slight changes. Therefore, Functor $\mathcal{F}_{\mathcal{A},f}$ is locally continuous and we can justify our minimization technique.

### 4.4  DIRECT DISTANCE COMPUTATION ON FUNCTIONAL SPACE

**Neural Tangent Kernel**  Two questions are left unanswered. 1. How can we implement this Functor $\mathcal{F}$? 2. How is the corresponding distance measure $d$ is defined?

We employ the Loss Tangent Kernel (LTK) (Chen et al., 2023) to define Functor $\mathcal{F}$. LTK $K$ integrates the loss derivative $\nabla_f \ell(f(x;\theta))$ to NTK $\Theta$ to measure the similarity between Functors $k(\cdot, x)$ and $k(\cdot, x')$ which is defined as follows:

$$K(x, x') \triangleq \langle k(\cdot, x), k(\cdot, x') \rangle = \nabla_f \ell(f(x))^\top \Theta \ \ell(f(x)) = \nabla_\theta \ell(f(x))^\top \nabla_\theta \ell(f(x'))$$

$$\nabla_f \ell(f(x)) \in \mathbb{R}^{C \times 1}, \ \nabla_\theta \ell(f(x)) \in \mathbb{R}^{P \times 1}, \ \Theta \in \mathbb{R}^{C \times C}, \ C = \#\text{classes}, P = \#\text{parameters}. \quad (7)$$

Replacing $\mathcal{F}_{\mathcal{A},f}(x)$ in Eq. (6) with $k(\cdot, x)$, we obtain

$$x_{t+1} = x_t - \nabla_{x_t} d(k(\cdot, x_t), k(\cdot, \mathcal{D}_s)) \quad (8)$$

$$\simeq x_t + \nabla_{x_t} K(x_t, \mathcal{D}_s) = x_t + \frac{1}{|\mathcal{D}_s|} \nabla_{x_t} \{ \nabla_\theta \ell(f(x_t; \theta))^T \sum_i^{|\mathcal{D}_s|} \nabla_\theta \ell(f(\zeta_i; \theta)) \} \quad (9)$$

$$\simeq x_t - \nabla_{x_t} d(\nabla_\theta \ell(f(x_t; \theta)), -\theta). \quad (10)$$

The above equations are induced in the following steps. First, as we mentioned in Sec. 3, $k(\cdot, x)$ exists in a functional space. Using this to replace $\mathcal{F}$ in Eq. (6) with $k(\cdot, x)$ we get Eq. (8). Then we approximate the distance term with the negative inner product in RKHS. Lastly, using that the loss gradient is aligned with the parameter itself (Ji & Telgarsky, 2020; Yun et al., 2021), we approximate the inner product between $x_t$ and $\mathcal{M}$ to the negative distance between the gradient of the loss and the negative of the trained model weight $\theta$ for easier calculation. However, in a practical setting, we cannot always obtain or use appropriate loss $\ell$ due to the following reasons: First, when designing the loss with labels, the generation is conditioned solely on the labels *i.e.*, we cannot implement unconditional generation. Second, in practical scenarios, the loss itself is often unknown.

**Augment-invariance loss**  We use the augmentation-invariance loss as a surrogate loss as it is an excellent estimate of the original loss for well-trained models. This is because well-trained models

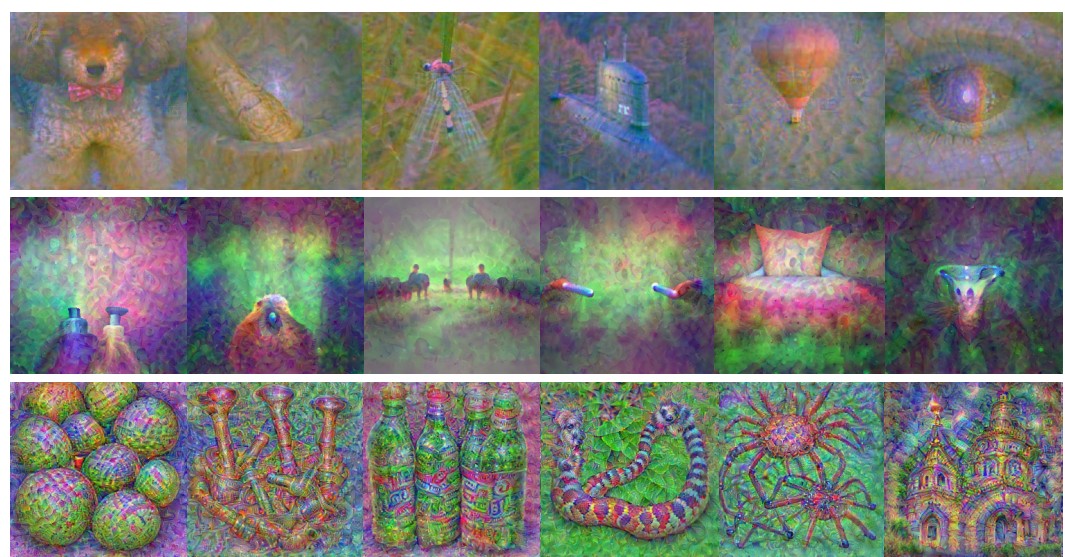

Figure 4: Examples of unconditionally generated images using various discriminative models. Top: DINOv2/LVD-124M (self-supervised learning). Middle: DeTR/COCO (object detection). Bottom: ResNet/Imagenet (image classification).

are known to produce similar outputs even with augmented inputs, expressed mathematically as $f(A(x)) \simeq f(x)$. This indicates that the loss dynamics strongly penalize augmentation-variant features. Therefore, using augmentation-invariance loss can serve as a surrogate loss:

$$\text{DFG:} \quad x_{t+1} = x_t - \nabla_x d(\nabla_\theta f(x_t; \theta) \| f(\mathcal{A}(x_t)) - f(x_t) \|, -\theta). \tag{11}$$

Now the score term can be calculated from a pretrained network, without having to approximate any distribution through learning. Here, $\| f(\mathcal{A}(x)) - f(x) \| := [\| f_1(\mathcal{A}(x)) - f_1(x) \|; \cdots; \| f_C(\mathcal{A}(x)) - f_C(x) \|]$ and from now on, we use $\lambda$ for $\| f(\mathcal{A}(x)) - f(x) \|$ for brevity. In our experiment, we set $d(a, b) = \| a/\|a\| - b/\|b\| \|$.

## 5 EXPERIMENTS

**Implementation Details**   In this paper, we utilized ResNet (He et al., 2015), DINOv2 (Oquab et al., 2024), and DeTR (Carion et al., 2020) models as the backbone for direct generation of $256 \times 256$ images. All models were obtained from the official repositories. DINOv2 is based on ViT (Vision Transformer) (Dosovitskiy et al., 2021) architecture and was trained on LVD-142M dataset. DeTR was trained using COCO dataset (Lin et al., 2014). For conditioning, we employed both ResNet and OpenCLIP (Ilharco et al., 2021) models. We used OpenCLIP as a base model. During the optimization process, we set gradient clipping value to 1e-5 and learning rate to 2 across all models. We calculate the weighted sum of the score term (DFG) and the conditional term for conditional generations. For ResNet the weight of the score term is set to 10, and the conditional term to 20. We also applied variance norm as a regularizer (Mahendran & Vedaldi, 2015). The regularlizer is added to Eq. (13) with its weight set to 20. Since the DFG size is proportional to the number of parameters, we adjusted the hyperparameters for the remaining experiments to match the size of the DFG loss.

### 5.1 UNCONDITIONAL GENERATION

What we obtain through Eq. (11) is the distance between an arbitrary point $x \in \mathbb{R}^{HW}$ and the image manifold $\mathcal{M}$. A question naturally arises. **Is DFG enough to generate images unconditionally from a discriminative model?** Can we approximate the distribution $p(x)$ of datasets, the fundamental objective of generative models?

The answer to the question turned out to be 'yes'. We conducted the demonstration as fairly as possible with different datasets, architectures and algorithms. For datasets, ImageNet, LVD-124M,

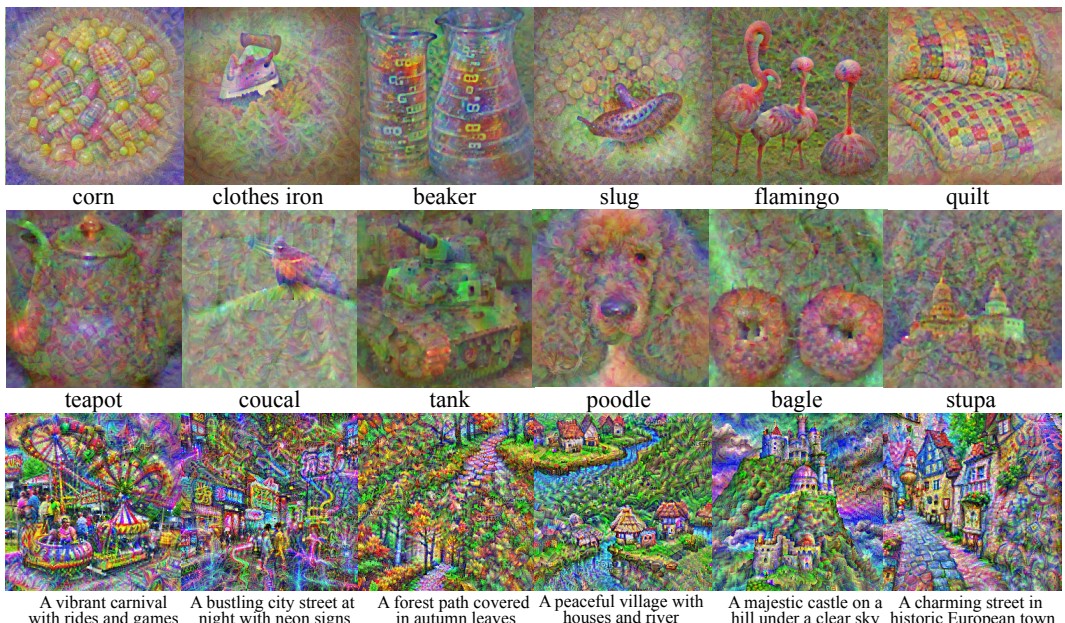

| corn | clothes iron | beaker | slug | flamingo | quilt |
| teapot | coucal | tank | poodle | bagle | stupa |
| A vibrant carnival with rides and games | A bustling city street at night with neon signs | A forest path covered in autumn leaves | A peaceful village with houses and river | A majestic castle on a hill under a clear sky | A charming street in historic European town |

Figure 5: Conditionally generated images using DFG. Top: Both the score model and the conditional model used ResNet/Imagenet. Middle: The score model used DINOv2/LVD-124M, and the conditional model used ResNet/Imagenet. Bottom: The score model used ResNet/Imagenet, and the conditional model used CLIP.

and COCO dataset were chosen. We selected three models of different architectures and algorithms: ResNet for ImageNet classification, DeTR for COCO object detection, and DINOv2 for self-supervised learning. We only utilized famous models from public repositories for transparency. No post-training was done because we are to show that generation is possible without modification or tuning for any arbitrary model.

For datasets, DFG successfully covered datasets of various aspects. DFG successfully generated images under ImageNet setting, which is considered the most common and general one at the moment. The high-quality image output using LVD-124M, which is one of the largest dataset used for vision tasks, proves that the generation happens regardless of the gigantic size of train dataset. Tests on COCO dataset that includes images of various objects, shapes and sizes in high-resolution implies that DFG can even handle high-resolution datasets. The results are favorable for different architectures and algorithms also. Fig. 4 shows the result of unconditional generation using three pairs of different models and algorithms. All three pairs successfully generated images of realistic shapes.

## 5.2 CONDITIONAL GENERATION

To generate images that we 'want', conditional term is added to Eq. (11) to guide generation. Conventional diffusion models create images with the following:

$$x_{t+1} = x_t + \underbrace{s_\theta(x_t, t)}_{\text{score}} + \underbrace{\nabla_x \log p_t(y|x_t)}_{\text{condition}}. \tag{12}$$

Noisy classifier $f_t$ is utilized instead of common classifier $f$ to implement the conditional term. This is because the classifier must be time-dependent and change with the time-step, as mentioned in Sec. 2. On the other hand, our method does not depend on time $t$, allowing us to implement the conditional term using naive classifiers, $\phi$. Our objective function can be written as below:

$$x_{t+1} = x_t \underbrace{- \nabla_x d(\nabla_\theta f(x_t; \theta)\lambda, -\theta)}_{\text{score}} + \underbrace{\nabla_x \log p_\phi(y|x_t)}_{\text{condition}}. \tag{13}$$

In Fig. 5 we used common pretrained classification model in a plug-and-play manner for generation. Our score and condition model for DFG successfully replaced the ones in the diffusion equation and

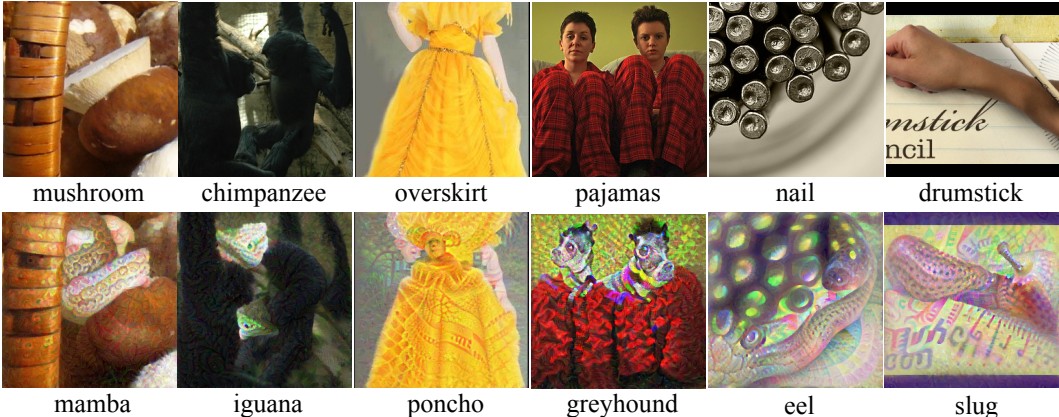

Figure 6: Examples of targeted editing, the top row contains original images while the bottom row shows editted images.

accomplished image generations. Furthermore, the last row of Fig. 5 suggests that not only classifier guidance, but also text-to-image (T2I) with a specific prompt input is possible when using CLIP.

## 6 APPLICATIONS

### 6.1 IMAGE EDITING

What a diffusion model learns is the reverse process of converting an image to pure noise. Based on this property, editing tasks in diffusion include sending the object image to a noise-like image and retrieving it back. Condition is injected into this process in the form of masking (Couairon et al., 2022). For the case of DFG, what it conducts is not the denoising process but the direct estimation of manifold distribution $p(x)$. Editing is therefore possible at image level without adding noise to the target image. Masking is also unnecessary. The result of directly applying conditional generation to real image without adding noise nor masking is shown in Fig. 6. The outstanding point is that the semantic has changed without largely deforming the input images. For example, in the photo of a chimpanzee, a spot of light fallen at the background is naturally transformed to a head of iguana. The body of the animal is trivially deformed to consist shadow and body of the iguana. This was possible by consistently injecting our score function as restraint, preventing the result from straying away from the manifold.

### 6.2 IMAGE INPAINTING

Image inpainting is a technique that visually restore the damaged or omitted part of the target image. To enable this, the ability to generate new contents according to the whole context it understood is mainly required. Diffusion model is suitable for inpainting tasks in this sense. Our methodology bears the same ability of solving inpainting tasks as diffusion models since it is, in the end, an algorithm that samples from noise. We generated an empty patch to mask the ground-truth and applied our DFG loss to it. Then the masked areas were effectively restored under the naive setting that only requires to meet the boundary condition. The results in Fig. 7 experimentally show that our algorithm shares the properties of diffusion models.

### 6.3 GLOBAL EXPLANATION

The core of our methodology boils down to suggesting a distance measure using the functor $\mathcal{F}_{\mathcal{A},f}$. A notable point is that we use a **conditioned** functor where $\mathcal{A}$ and model $f$ is fixed. This implies that DFG moves $x$ to the center of the manifold $\mathcal{M}$ with the lens of the model $f^\star$ learnt from $\mathcal{M}$. It in itself plays a role of eXplainalbe AI (XAI), especially feature visualization, as it tries to understand the fundamentals of a model using the corresponding parameter only. DFG takes a step further than existing algorithms. Conventonal algorithm aim to visualize using gradient descent only. However,

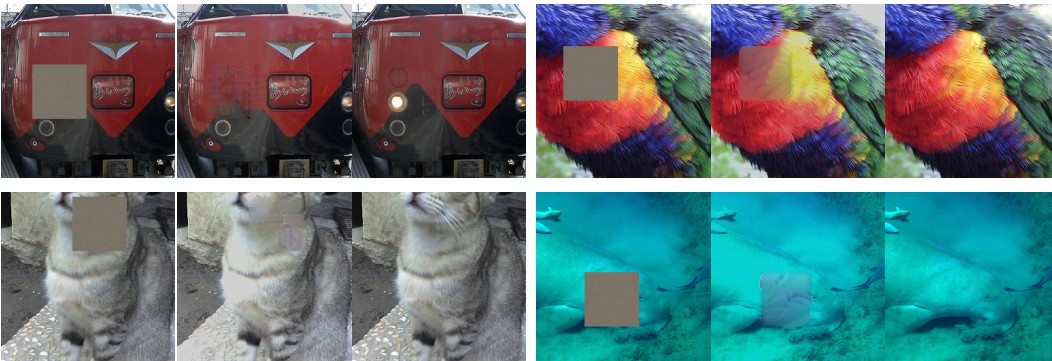

Figure 7: Examples of the impainting task. For each set, the left is the inpainted image, the middle is the reconstructed one, and the right is the original image.

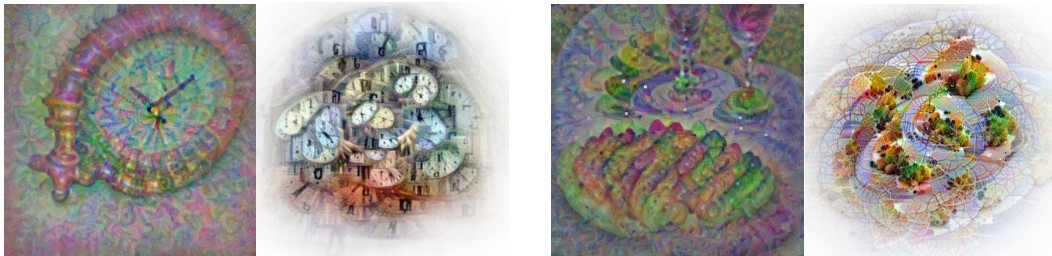

Figure 8: Comparison between DFG (left) and existing XAI method (Fel et al., 2023) (right).

just because the generated images can be classified to intended classes, we cannot insist that it actually represents the learned criterion. For example, adversarial attack also updates images with the method called gradient ascent, but no one argues can be made regarding that the process actually provides a new decision criterion. Moreover, as Fig. 8 reveals, conventional feature extraction algorithm generates outputs by exaggerating characteristics of objects. The resulting images are far from realistic. Our DFG method offers mathematical justification on how the generated image reflects the manifold, and generates realistic images.

The result showing that DFG can generate high fidelity images offers more detailed explanation than the feature extraction method conventionally used to explain deep learning. Fig. 8 is a good example. On the left, the clock generated with DFG points to 10:10 and 35 seconds. This leads to the intuition that large group of clock data in the train set may have displayed that certain time. Actually, the very time the generated clock points to is what many watch advertisements adopt in order to emphasize the logo. On the right is the visualization of the class 'plate'. One can see that wineglasses have been generated alongside the intended plate. The result may indicate that plates are paired to glasses of drinks for many cases in the dataset. Conventional method cannot induce the same conclusion.

## 7 CONCLUSION

In this paper, we demonstrate for the first time that discriminative models can be used as general generative models, and introduce a schema called Discrimination for Generation (DFG) to achieve this. Using the discrimination model and the Neural Tangent Kernel, DFG directly calculates the distance from a data point to the data manifold and generates images based on that measure. We validated DFG through extensive experiments, showing that it can generate images in an architecture- and dataset-agnostic manner, with or without conditions. Furthermore, DFG can be applied to other downstream tasks. It can be used not only for conventional vision challenges like image editing and inpainting, but also for global explanations leveraging the concept of DFG.

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
