

Figure 9: More images generated by ResNet50/Imagenet

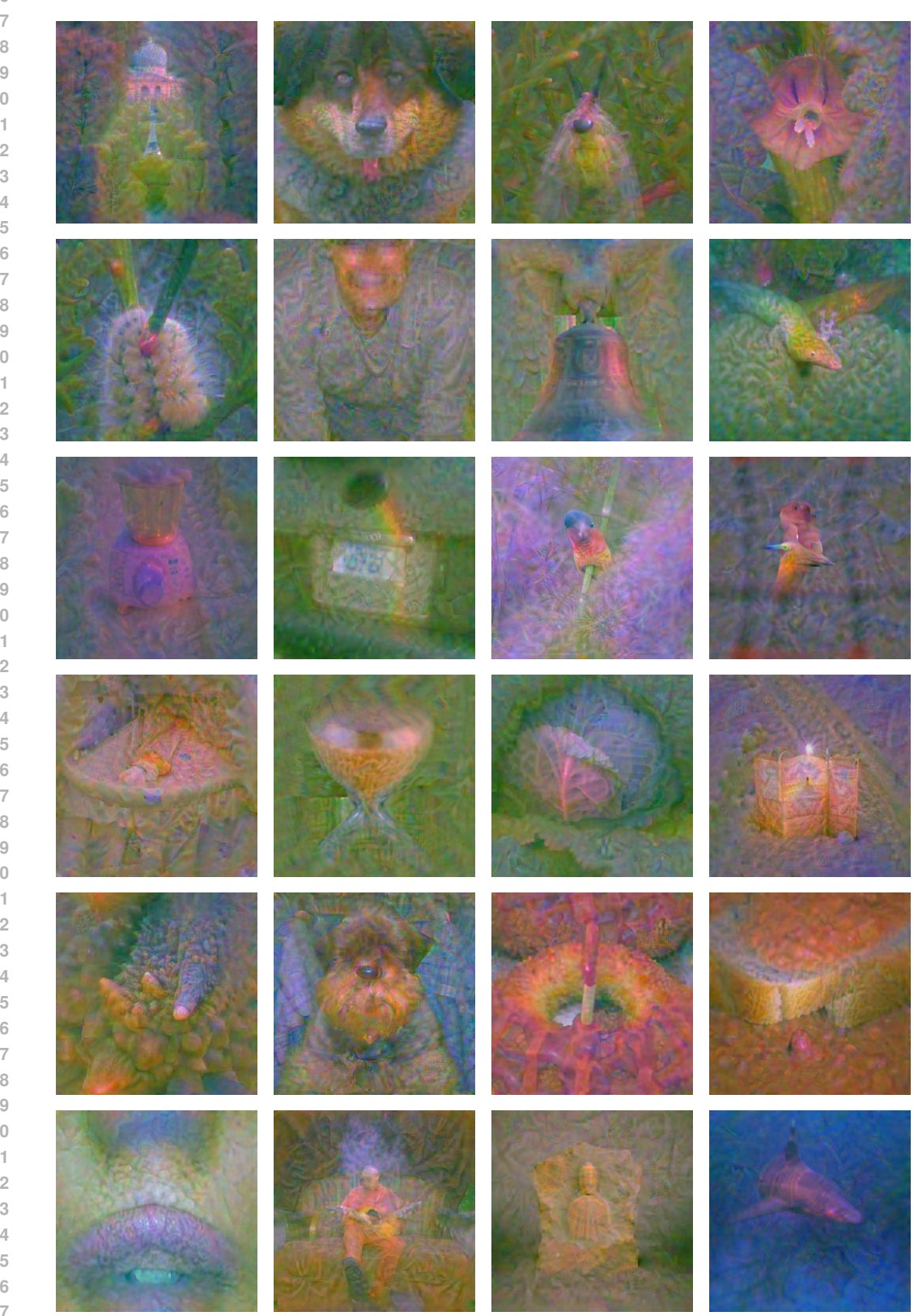

Figure 10: More images generated by DINOv2/LVD-124M

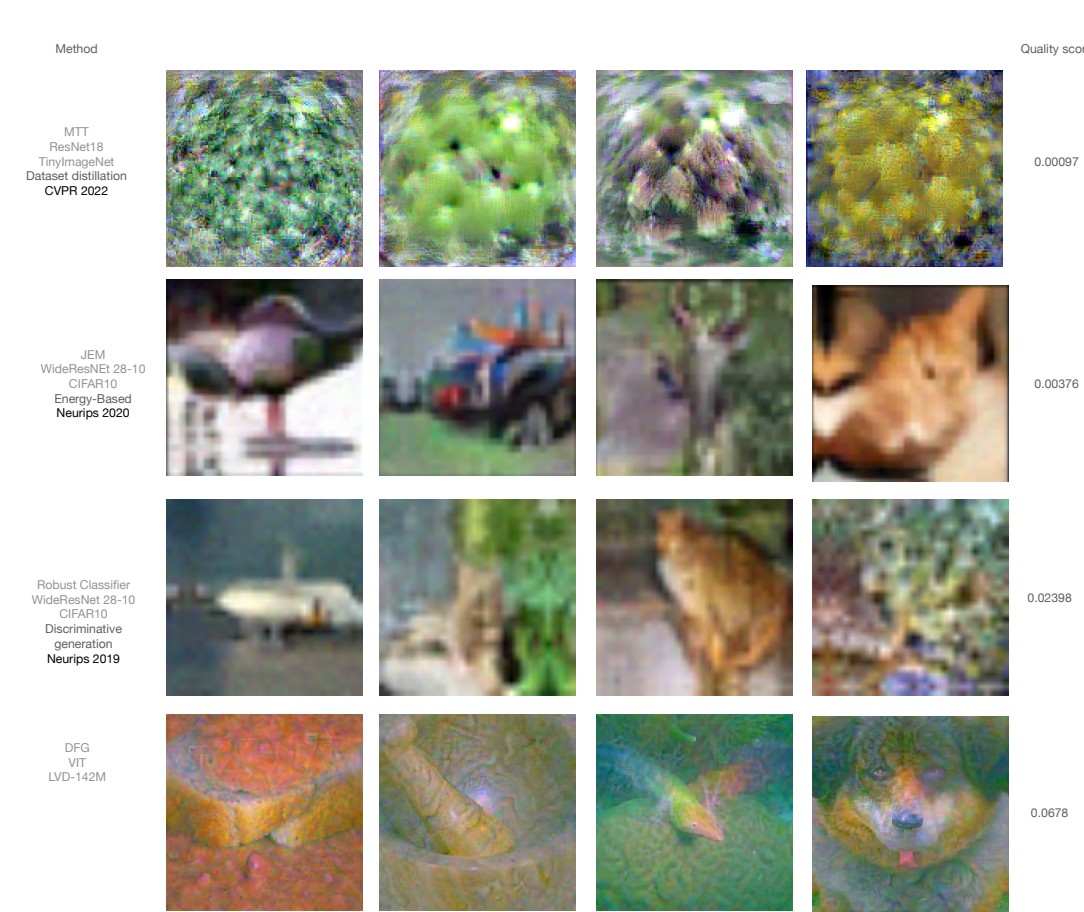

Figure 11: Comparison between other benchmarks and DFG (ours). The quality metric is derived from Q-align, which allows for a unified quality score to be measured across different datasets. In the second row, JEM represents a generative model, while the remaining rows correspond to discriminative models.

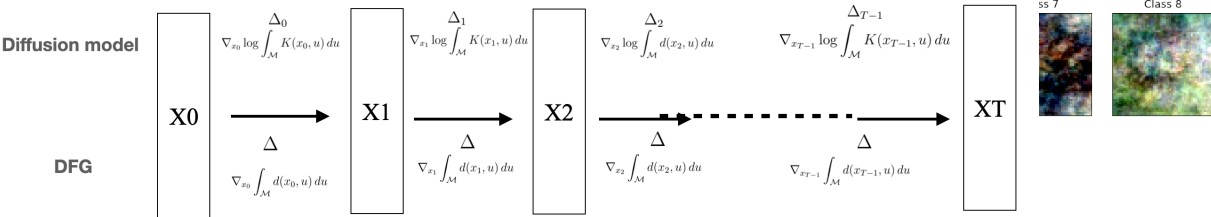

Figure 12: comparison between diffusion model and DFG. Both methods actually solves same procedure. As the probability can be seen as the limite case of Kernel Density estimation, in can be write as $\nabla_x \log \int_{\mathcal{M}} K(x, u) du$

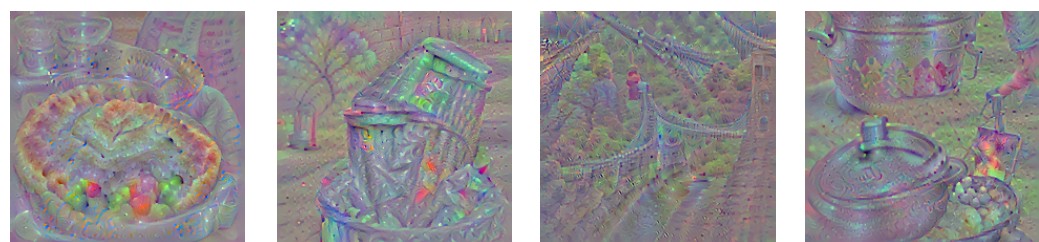

X0  X1  X2  XT

With seed

Class 0  Class 1  Class 2  Class 3  Class 4  Class 5  Class 6  Class 7  Class 8

Figure ... cept
izati... ne...
(Bot... ple...

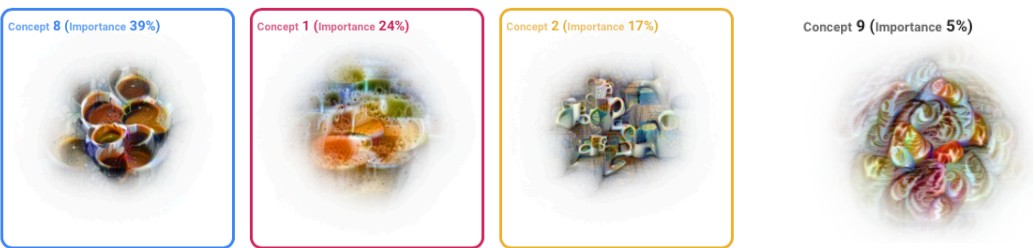

Figure 14: An example of different weighting function. The photos are generated by cross-entropy weighting function.

Concept 8 (Importance 39%)  Concept 1 (Importance 24%)  Concept 2 (Importance 17%)  Concept 9 (Importance 5%)

Figure 15: An example of Feature Visualization. The figure is from https://serre-lab.github.io/Lens/