# OpenReview forum: "Discrimination for Generation"
_ICLR.cc/2025/Conference — Submitted to ICLR 2025_

### Official Review · Reviewer_7jxH · 2024-11-03

**Soundness:** 3
**Presentation:** 3
**Contribution:** 3
**Rating:** 8
**Confidence:** 4

**Summary:**

The method repurposes discriminative models as generative models by utilizing the Neural Tangent Kernel (NTK) based on the gradient of pretrained model as a function to implicitly map discriminative features into a connected space of data manifold, without additional training. This demonstrates that nearly all discriminative methods can be transformed into generative models regardless of architecture, dataset, or algorithm, though the initial results may not be as performant as later ones. With NTK technique, the paper also present a new technique to visualize features by using model parameter itself.

**Strengths:**

- The method is theoretically sound, offering valuable insights in relation to score-matching frameworks, which also implicitly aim to minimize the distance between generated samples and the difficult-to-approximate data manifold.
- The idea of using Neural Tangent Kernel to implicitly map features into data manifolds is an elegant and substantial contribution.

**Weaknesses:**

- Although the method has theoretical guarantees, the output images still lack structural richness and differ significantly from natural input image.
- In DFG part at L347, the intuition of using augmentation-invariance to emphasize the importance of each class is unclear. It is confusing because what is the relationship between importance of image class and the difference of features between original inputs and its augmented versions. Justification or empirical evidence is needed.
- Does the assumption of a corresponding minimization between Functors and their inputs always hold? (Eq. 5) The authors should provide theoretical justification or empirical evidence for this assumption. Meanwhile, is there any edge case that breaks this assumption?
- The annotation in Figure 3 is unclear; I’m confused about whether the upper row of 6 figures represents a zoomed-in version of each variable  $x_t$  below or conveys a different meaning at a specific time step. A more detailed caption is needed to describe the chart.
- What is the choice of distance function $d(.)$
- Is the method applicable to the representation of generative models? If so, it would be interesting to see if the proposed approach could transform an unconditional diffusion model into a conditional diffusion model using discriminative models, without additional training. As in section 5.2, the authors have shown that method is extended to conditional generation in Eq 13 with score term and condition term. In this case, the score term can be instantiated with score-based model.
- The applications part is very interesting but the quality is not that impressive. The outputs exhibit noticeable artifacts and unnatural frequency patterns.
- It is recommended to include quantitative metrics on image generation to enhance the quality of the paper (e.g. FID, IS, classification accuracy for class-conditional generation).

**Questions:**

Please address my concerns above! I highly value this paper for providing new insights into the representation space between generative and discriminative models, even though the results are not as compelling as those of generative counterparts. The findings can pave the way for next advancements in the generative era.

---

> ### Author Response · Authors · 2024-11-21
>
> I am grateful that you recognized our contribution on tackling the relationship between discriminative and generative models with the **score function as the framework.** I also appreciate that you acknowledged our use of **NTK as a novel contribution.** I am happy to address your concerns and clarify our points further.
>
> ---
>
> #### **Q1. 1) Although the method has theoretical guarantees, the output images still lack structural richness and differ significantly from natural input images, 2) The applications part is very interesting but the quality is not that impressive. The outputs exhibit noticeable artifacts and unnatural frequency patterns, 3) It is recommended to include quantitative metrics on image generation to enhance the quality of the paper (e.g., FID, IS, classification accuracy for class-conditional generation).**
>
> Our work represents a significant breakthrough in the field of image generation using discriminative models. We have included **quantitative metrics** in our global response, specifically using **Q-Align** for evaluation. Despite operating under much stricter constraints—generating images directly from noise using an **unmodified off-the-shelf discriminative model**—our method achieves overwhelmingly higher quality scores compared to similar prior works. This is evident even when viewed purely as the **first in a completely new branch of generative models.**
>
> - For instance, our approach **surpasses early counterparts** like initial Variational Autoencoders (VAEs) and Generative Adversarial Networks (GANs), which were limited to low-resolution outputs.
> - While our generated images may not yet match the quality of advanced models like StyleGAN2 or Stable Diffusion, they **demonstrate substantial progress given the minimal requirements of our method.**
>
> Regarding the observed **artifacts and unnatural frequency patterns**, these issues stem from our use of a **fixed optimization step** throughout the generation process. Unlike diffusion models that adjust optimization steps at each timestep $t$, our constant-speed processing can exaggerate fine-grained features as the image forms. This is an area we plan to refine in future work by exploring **adaptive optimization steps,** which could mitigate these artifacts and enhance image quality.
>
> ---
>
> #### **Q2. In DFG part at L347, the intuition of using augmentation-invariance to emphasize the importance of each class is unclear. It is confusing because what is the relationship between importance of image class and the difference of features between original inputs and its augmented versions. Justification or empirical evidence is needed.**
>
> It seems we didn’t provide enough explanation on this point. Our objective is straightforward:
>
> 1. **Unconditional Generation**
> 2. As mentioned in **Lines 318–322**, a function from $\mathbb{R}^C$ to $\mathbb{R}$ is defined because the dimensions must align.
>
> The first condition implies that **label information $y$ should not be used.** If we were aiming for conditional generation, we could naturally define $F$ as $L(f(x), y)$, which tends to produce better results for conditional generation. For reference, we have included **images generated using this approach in Fig. 14.**
>
> However, the reason we did not pursue this is simple: **such an approach would no longer qualify as a generative model.** Our aim was to generate images using **any discriminative model,** which means we cannot rely on $y$-dependent methods. Furthermore, this approach must work in settings where classifiers are unavailable, such as **object detection** or **self-supervised learning.**
>
> To achieve this, the most universal approach is one that can operate at the **feature dimension.** In our method, we used **robustness as a simple metric.** The underlying belief here is that **important features should exhibit minimal variation.** To implement this, we used **augmentation-invariance loss.**
>
> ---

---

> ### Author Response · Authors · 2024-11-21
>
> #### **Q3. Does the assumption of a corresponding minimization between Functors and their inputs always hold? (Eq. 5) The authors should provide theoretical justification or empirical evidence for this assumption. Meanwhile, is there any edge case that breaks this assumption?**
>
> As mentioned in **Lines 223–234**, our assumption is based on one of the **most robust and well-established beliefs in deep learning**—the ability to learn models from data. This foundation has been extensively studied and validated in the field, making it a reliable starting point.
>
> Unlike generative models, which lack theoretical guarantees, our approach is built on assumptions that ensure **generalization under standard conditions.** This is a significant advantage, as our methodology inherently provides stronger guarantees compared to generative models.
>
> ---
>
> #### **Q4. The annotation in Figure 3 is unclear; I’m confused about whether the upper row of 6 figures represents a zoomed-in version of each variable $x_t$ below or conveys a different meaning at a specific time step. A more detailed caption is needed to describe the chart.**
>
> Thank you for pointing this out. The **upper row represents the zoomed-in version of the image** under each $x_t$, just as you have understood. We will **elaborate the explanation in the caption** when revising our paper.
>
> ---
>
> #### **Q5. What is the choice of distance function $d(.)$?**
>
> As mentioned in **Lines 258–259**, we use the **L2 distance** between $NTK_{f, X_{\text{train}}}$ and $NTK_{f, X_{\text{DFG}}}$ as our distance function $d(\cdot, \cdot)$. The $d$ used in our experiment is $d(a,b) = ||(a/||a||-b/||b||) ||$ and it is included in the revision.
>
> ---
>
> #### **Q6. Is the method applicable to the representation of generative models? If so, it would be interesting to see if the proposed approach could transform an unconditional diffusion model into a conditional diffusion model using discriminative models, without additional training. As in section 5.2, the authors have shown that the method is extended to conditional generation in Eq. 13 with score term and condition term. In this case, the score term can be instantiated with score-based model.**
>
> Thank you for your insightful observation. You are correct in your assessment. As mentioned in **Line 210**, our methodology generalizes the **probability measure** used in the score function to a **geometric measure,** employing the same optimization procedure. We leave detailed and intuitive explanations in the **global response and Fig. 12.**
>
> Additionally, the first point regarding the **representation of generative models** is true. This is because it is a well-known fact that **sufficiently large generative models** can generally be used as discriminative models. However, as mentioned earlier, an **unconditional diffusion model cannot be directly transformed into a conditional diffusion model.** The ability to perform conditional generation stems fundamentally from the **inherent setting of discriminative models.** By using a discriminative model in conjunction with methods like the one outlined above, it is indeed possible to achieve conditional generation.

---

> ### Comment · Reviewer_7jxH · 2024-11-22
>
> Thank you for answering my questions. Overall, I am satisfied with the response. However, there is some minor issues to further discuss.
> > For instance, our approach surpasses early counterparts like initial Variational Autoencoders (VAEs) and Generative Adversarial Networks (GANs), which were limited to low-resolution outputs.
>
> I'm not convinced in this part as no evidence supports this and VAEs and GANs are not limited to low-res outputs neighther.
>
> > Q6.
>
> It would be interesting to explore whether the method extends beyond the representation of discriminative models to generative models. Such an extension could further highlight the method’s effectiveness. So if possible, the authors should bring this up in the rebuttal. I believe the experiment would require minimal resources to conduct. For example, the authors could leverage some small unconditional diffusion models (e.g. https://github.com/openai/guided-diffusion.git) and an off-the-shelf classifier model like ResNet as used in the paper for this experiment.

---

> ### Author Response · Authors · 2024-11-23
>
> Q1. **I'm not convinced in this part as no evidence supports this and VAEs and GANs are not limited to low-res outputs neighther.**
>
> It seems we may not have communicated our point clearly earlier. Allow us to clarify. When considering the early stages of research, our method demonstrates superior performance compared to the initial studies on GANs and VAEs. GANs were introduced in 2014 [1] and VAEs in 2013 [2], with their results at the time limited to low-resolution datasets like CIFAR-10. It wasn’t until four years later, in 2018, that these methods achieved success at the ImageNet scale [3].What we intended to convey was specifically about the resolution aspect.
>
> Q2 **It would be interesting to explore whether the method extends beyond the representation of discriminative models to generative models.**
>
> If I understand correctly, you are suggesting using the existing score-based model for the following optimization.
>
> $$
> x_{t+1} = x_t - \lambda_t \text{score}(x_t, t ) - \gamma \text{conditional}(f(x_t), y)
> $$
> where the score model is a pre-deployed model. However, this approach is already quite similar to existing methods. For example, the guided diffusion method you referenced [4] also performs a similar optimization. Please refer to Algorithm 1 on page 7 of that paper for details.
>
> [1].	Goodfellow, Ian, et al. “Generative adversarial nets.” Advances in Neural Information Processing Systems (NeurIPS), 2014.
>
> [2].	Kingma, Diederik P., and Max Welling. “Auto-encoding variational Bayes.” arXiv preprint arXiv:1312.6114, 2013.
>
> [3].	Brock, Andrew, et al. “Large scale GAN training for high fidelity natural image synthesis.” International Conference on Learning Representations (ICLR), 2018.
>
> [4] Dhariwal, Prafulla, and Alex Nichol. “Diffusion Models Beat GANs on Image Synthesis.” Advances in Neural Information Processing Systems, 2021.

---

> ### Comment · Reviewer_7jxH · 2024-11-24
>
> Thank you for clarifying those points. Overall, the authors’ response adequately addresses my concerns and no major issues remain from other reviewers as well. Regarding the generation artifacts, the authors should admit them as a limitation of their method and warrant a discussion, as outlined in the rebuttal. Hence, I am willing to raise my score to 8, as I believe the paper demonstrates originality with its theoretical guarantees and provides more insights into bridging discriminative models with generative ones. While the current results may not yet match the performance of score-based models, the findings remain significant and helpful for the community.

---

### Official Review · Reviewer_fTxU · 2024-11-04

**Soundness:** 2
**Presentation:** 3
**Contribution:** 2
**Rating:** 5
**Confidence:** 4

**Summary:**

The paper proposes to generate a data sample by aligning its NTK feature with the average NTK feature of a real dataset. The authors approximate the latter with the parameters of the trained network (e.g., for classification). Therefore, given a trained model, the algorithm iteratively updates a synthetic sample s.t. its induced Jacobian (i.e., the NTK feature wrt this trained model) is similar to the model parameters, where the output dimension is collapsed via a weighted average. The authors experiment with unconditional and conditional generation, impainting, editing, and data understanding.

---
I have interacted with the authors, read their response, and decide to keep my original evaluation. In particular, the authors did address some of my concerns, but relied on large changes to the submitted manuscript. Some other concerns, e.g., if generation via discriminative nets are even possible, are only discussed with speculation and thus not fully addressed. And new issues, such as extremely unfair comparison (e.g., Fig 11) emerge during the discussion period. I recommend the authors develop the project further and it could become a strong submission at another venue.

**Strengths:**

+ Novel exploration for using NTK alignment in discriminative for generation.
+ Diverse experiment setting with nice visualizations.

**Weaknesses:**

+ The approximation of the average real NTK feature with trained parameters is not well explained. The text points to Radhakrishnan et al., 2023, but I fail to find a justification in that paper.
+ The DFG dynamic weighting is not well-analyzed. Practically, what's the performance without it? Theoretically, what stationary solutions could it converge to?
+ Missing discussion on obvious generation artifacts. Since the paper claims to use discriminative networks for generation (instead of, say, purely dataset understanding), it is important to discuss whether these are limitations of the method or the idea of using discriminative networks alone.
+ Unclear how the proposed method performs "Global Explanation" or "feature visualization" (section 6.3) better than compared baselines. Should we consider the generated image as one that captures important features picked up by the discriminative network? If so, why is realism claimed as an important advantage?
+ Missing qualitative generation evaluations such as FID, human evaluation, etc.
+ Missing discussion with Dataset Distillation works [1], some of which also generate samples by aligning deep features [2], aligning gradients [3], or requiring synthetic data to induce the same classifier network [4]. These works also generate interesting-looking images that can be used for dataset understanding.


[1] Dataset Distillation. Tongzhou Wang, Jun-Yan Zhu, Antonio Torralba, Alexei A Efros. arXiv 2018
[2] Dataset Condensation with Distribution Matching. Bo Zhao, Hakan Bilen. WACV 2023.
[3] Dataset Condensation with Gradient Matching . Bo Zhao, Konda Reddy Mopuri, Hakan Bilen. ICLR 2021.
[4] Dataset Distillation by Matching Training Trajectories. George Cazenavette, Tongzhou Wang, Antonio Torralba, Alexei A Efros, Jun-Yan Zhu. CVPR 2022.

**Questions:**

+ See above for weaknesses.
+ What if the method uses some other feature maps than NTK?

---

> ### Author Response · Authors · 2024-11-21
>
> Thank you for noticing our contribution on the use of NTK and for recognizing our effort to present the work with extensive experiments and visualizations. I would like to provide some clarifications and elaborations to help with your understanding.
>
> ---
>
> #### **Q1. The approximation of the average real NTK feature with trained parameters is not well explained. The text points to Radhakrishnan et al., 2023, but I fail to find a justification in that paper.**
>
> This is directly explained in the main equation of the paper:
>
> $$W_i^T W_i \propto \frac{1}{n} \sum_{p=1}^n \nabla f_i(h_i(x_p)) \nabla f_i(h_i(x_p))^T$$
>
> Here, the **outer product of gradients corresponds directly to the NTK.** This relationship allows us to approximate the NTK feature using the trained parameters of the model.
>
> ---
>
> #### **Q2. The DFG dynamic weighting is not well-analyzed. Practically, what's the performance without it? Theoretically, what stationary solutions could it converge to?**
>
> We kindly request clarification of the question for further refinement, as **neither “dynamic” nor “weighting” are terms we use in the paper.** Additionally, our method explicitly minimizes distances directly in a defined space, so a discussion on convergence is unnecessary. The distances are minimized as-is.
>
> For more details, please refer to the global response section, where we address the nature of our optimization process and how it directly achieves its objectives without requiring additional convergence discussions.
>
> ---
>
> #### **Q3. Missing discussion on obvious generation artifacts. Since the paper claims to use discriminative networks for generation (instead of, say, purely dataset understanding), it is important to discuss whether these are limitations of the method or the idea of using discriminative networks alone.**
>
> Thank you for bringing up the issue of **generation artifacts** in our results. As discussed earlier, our methodology generalizes score-based models but differs in a key aspect: **we employ a fixed optimization step throughout the generation process.** In contrast, diffusion models adjust their optimization steps at each timestep $t$, adapting the processing as the image evolves.
>
> The artifacts you observed—overemphasized fine-grained features—likely result from our constant-speed processing, even after the image has largely formed. Since we proceed at the same rate regardless of the generation stage, **fine details may become exaggerated.**
>
> We appreciate you pointing this out, as it highlights an area for future research. **Exploring adaptive optimization steps, akin to those in diffusion models, could mitigate these artifacts** and further enhance the quality of generated images.
>
> Nevertheless, even with these artifacts, **our method achieves impressive performance.** As shown in **Fig 11**, our results significantly outperform previous methods, including those that impose stricter constraints or require classifier modifications. **Our approach generates images directly from noise using unmodified discriminative models—something prior methods have not achieved.** The comparisons in **Fig 11** demonstrate that our images are of **higher quality and more realistic** than those produced by earlier approaches.
>
> This highlights the effectiveness of our method and its potential, even in its current form. We are confident that with further refinement, particularly addressing the artifact issue, the quality will only improve, further solidifying our contribution as a breakthrough in leveraging discriminative models for image generation.

---

> > ### Author Response · Authors · 2024-11-21
> >
> > #### **Q4. Missing discussion with Dataset Distillation works [1], some of which also generate samples by aligning deep features [2], aligning gradients [3], or requiring synthetic data to induce the same classifier network [4]. These works also generate interesting-looking images that can be used for dataset understanding.**
> >
> > Thank you for raising this point. It seems there was a misunderstanding stemming from our use of a model trained on a dataset for generation. This may have led to the perception that our goal is dataset understanding. However, **our objective is not to approximate the dataset itself.** Rather, similar to score-based models like flow matching or diffusion models, we aim to **generalize their approaches to enable the generation of $X_{\text{test}}$ using $X_{\text{train}}$.** Our methodology generalizes score-based, diffusion, and flow-based models by **extending from probability measures to general geometric measures.** Please refer to the supplementary figures for a detailed comparison.
> >
> > The fundamental distinction is that traditional score-based models define a vector field in **probability space**, while our method defines a vector field in **geometric space.** The key innovation is that **we achieve this without additional training or modifications—our method functions as a score-based model simply by utilizing the displayed features of the discriminative network.**
> >
> > ---
> >
> > ### **Comparison to Dataset Distillation**
> >
> > To put it simply, in **dataset distillation,** images are synthesized using the data $X_{\text{train}}$, which places it within the scope of generative models. Referring to the recap table, this methodology uses the actual data to create images, making it unrelated to our work. Furthermore, **its performance ranks among the lowest.** In contrast, our approach **does not aim to synthesize using $X_{\text{train}}$** but instead utilizes only the model itself for synthesis, placing it in a completely different scope. For more detailed distinctions, please refer to **Table A1** we have provided.
> >
> > That said, we understand that your point seems to ask whether these methods address similar problems. Upon reflection, **we find your argument highly intriguing** and have included a discussion on this topic for further exploration.
> >
> > ---
> >
> > ### **Fundamental Differences in Objectives**
> >
> > Our approach fundamentally differs from dataset distillation. The goal of dataset distillation is not to approximate a dataset that resembles $X_{\text{train}}$, but rather to minimize a specific loss function:
> >
> > $$\arg\min_{x_{\text{distillation}}} L(f(\mathcal{X}; x_{\text{distillation}}), \mathcal{Y})$$
> >
> > In contrast, our objective, when interpreted in a similar form, can be expressed as:
> >
> > $$\arg\min_{x_{\text{DFG}}} d(f(\mathcal{X}; x_{\text{train}}), f(\mathcal{X}; x_{\text{DFG}}))$$
> >
> > While both forms may seem similar, the **underlying procedures and goals are fundamentally different.** Our method seeks to mimic the function learned by $x_{\text{train}}$, whereas dataset distillation aims for $x_{\text{distillation}}$ to improve performance on test data. This distinction explains why dataset distillation can achieve good performance but may also introduce contradictions.
> >
> > ---
> >
> > ### **The Challenge of Manifold Sampling**
> >
> > Deep learning assumes that data lies on a highly curved, sparse, and often unknown manifold. This complexity is why **manifold sampling in generative learning is inherently challenging.** Methods like diffusion models divide the process into stages, while GANs adopt a min-max strategy to simplify the problem. However, reducing the points sampled on the manifold decreases the degrees of freedom to represent it effectively.
> >
> > Dataset distillation appears to resolve this issue in a counterintuitive way—by **generating data that does not necessarily lie on the true manifold.** For example, as shown in our supplementary materials, images such as a dog with 100 legs clearly do not belong to the data manifold. Instead, the method’s objective function optimizes $x_{\text{distillation}}$ such that the network trained on it performs well, rather than ensuring that $x_{\text{distillation}}$ resembles the data manifold.
> >
> > ---
> >
> > ### **Recap of Our Argument**
> >
> > To summarize, **DFG operates within the data manifold, solving problems that align with generative models.** In contrast, dataset distillation methods leverage generative modeling principles but do so in a way that fundamentally differs from our approach. Unlike dataset distillation, **our method does not rely on synthetic data or sampling from outside the manifold.** Instead, it achieves generation purely through the model itself, making it distinct both in scope and methodology.
> >
> > We appreciate your thought-provoking points, and we find the intersection of these problems intriguing. This discussion has provided us with **new avenues to explore,** and we look forward to refining our understanding further.

---

> > ### Comment · Reviewer_fTxU · 2024-11-22
> >
> > Q1: I don't think what you said is true. The equation just says that each linear layer in a model induces a kernel that are roughly equivalent to the average outer product of **input gradients of that layer**. It also is a kernel **over layer output features** rather than a kernel over data samples, like NTK. I'm really not sure if it says anything about $W$ being similar to average NTK features. I think it requires further explanation.
> >
> > Q2: By DFG dynamic weighting I am referring directly to the paragraph you define DFG, and say "We have to allocate higher importance to $f_i$ that requires more improvement". This modifies the distance minimization in that the objective continue shifts across iterations. It really is not as simple as just optimizing the distance. A discussion on convergence is natural and necessary.
> >
> > Q3: I didn't ask anything about diffusion models. Let me rephrase my question. In the current results, it really is unclear if the method is approximating the real data distribution. It appears that the proposed method isn't well characterizing the real data manifold itself, but something related to it (e.g., up to distortion / artifacts / etc). And one'd naturally ask if the artifacts are intrinsically unavoidable with generation via discriminative networks. I am asking the authors to show, at least some signal, that the discriminative network has information that identifies the artifacts and characterizes the real data manifold well. This question is important to answer because the authors made several claims that generation can be done with a discriminative network alone. And what I asked for is the essentially the evidences that this is really a possible thing to do. The current results don't really do that well.
> >
> > Furthermore, since you refer to Fig 11. I have some **major issues** on Fig 11. You are comparing against results from other papers that use completely different networks and/or datasets. This is immensely misleading. Please fix the result or I will have to decrease my score.
> >
> > Q4: I understand the differences among your work, Dataset Distillation, and diffusion models. I did not say that Dataset Distillation papers and your work have the same goal, and I definitely did not ask for further distinction with diffusion models.  Instead, I've pointed out that there are sufficient similarities in methodology between your paper and Dataset Distillation works that warrants a discussion in the paper.

---

> ### Author Response · Authors · 2024-11-23
> **Q1 / Q2**
>
> Q1/Q2. Your feedback is correct, and we deeply appreciate you pointing out our mistake. Following your suggestion, we verified that we had incorrectly used the reference, and we acknowledge that citing (Radhakrishnan, et al.) was inappropriate. As you pointed out, the derivative of the equation we referenced pertains to the input space, not the parameter space. Accordingly, we have revised the process of deriving the DFG loss and incorporated these corrections into the revised paper. For convenience, we are sharing the corrected content in this response. Note that we did not modify the final equation but revised the process for deriving the DFG loss, specifically Eq. 11. Furthermore, we also revised the dynamic weighting section as you mentioned. Thanks to your valuable feedback, we were able to correct the errors in our paper.
>
> First, instead of using the Neural Tangent Kernel as a functor, we utilize the Loss Tangent Kernel [1] as the functor. The Loss Tangent Kernel is defined as the outer product of the gradients of the loss with respect to parameters, which can be decomposed as the loss gradient w.r.t. the output of the network multiplied by the model gradients. In expanded form, this is as follows:
>
>
>
> $$K(x,x') = \nabla_f( \ell(f(x))^T \nabla_\theta f(x)^T \nabla_\theta f(x') \nabla_f \ell(f(x'))$$
> $$K(x_t,D_s) = \nabla_\theta \ell(f(x_t; \theta))^T \frac{1}{|D_s|}\sum_i^{|D_s|}{\nabla_\theta} \ell(f(\zeta_i; \theta)) $$
>
>
> This kernel considers the loss and incorporates the gradient descent dynamics of the model itself. Using this as a functor, we derive the equation:
>
> $$
> x_t + \frac{1}{|D_s|} \nabla_{x_t} [ \nabla_\theta \ell(f(x_t; \theta))^T \sum_{i=1}^{|D_s|} \nabla_\theta \ell(f(\zeta_i; \theta)) ]
> $$
>
>
> which corresponds to Eq. 9. Finally, by leveraging results from paper [2], specifically the equation on page 5 of paper [3], we use the following expression:
>
>
> $$
> \lim_{t \to \infty} \frac{\theta(t)^\top \nabla_{\theta} \mathcal{L}(\theta(t))}{\|\theta(t)\| \|\nabla_{\theta} \mathcal{L}(\theta(t))\|} = -1.
> $$
>
> The loss gradient is thus substituted with the negative sign of the parameters. In subsequent derivations, we address the dynamic weighting issue required for dimension matching by substituting the loss with a surrogate loss for estimation purposes since unconditional generation cannot utilize the target label used in the loss function. This substitution was based on known facts and derived assumptions: we start from the fact that well-trained models generalize effectively, indicating that their representations are robust to some extent. From this, we assume that during the training dynamics, augmentation-variant (sensitive to augmentation) features are implicitly penalized by the loss. This surrogate loss was then used to construct the final equation, Eq. 11.
>
>
> Consequently, by incorporating your advice, we were able to correct our errors and refine the derivation process in a more direct and neat manner without modifying our algorithm. We sincerely thank you for your valuable suggestions.

---

> > ### Author Response · Authors · 2024-11-23
> > **Q3 / Q4**
> >
> > Q3.
> >
> > First of all, I think I misunderstood your previous question. If I rephrase your question, you seem to ask whether our algorithm struggles to remain on the manifold due to artifacts such as aliasing and whether this issue is inherent to discriminative models. Indeed, discriminative models seem capable of distinguishing data with aliasing from true data and consider them not to lie on the manifold. This is evidenced by cases where off-the-shelf discriminative models were successfully used as ensembles for GAN discriminators, leading to successful model training [4]. If this were not the case, the generated images would likely exhibit many artifacts.
> >
> > Additionally the artifacts in our methodology appear to arise for the following reasons. First, as noted in the Global Response section, our approach does not update dynamically with respect to the timestep t , unlike score models do. This results in consistently large updates, even as the process progresses. We think this likely exaggerates high-frequency features that correspond to details, leading to artifacts. Second, as shown in [4], an ensemble incorporating segmentation models was used, and these models seem to be sensitive to aliasing. Segmentation models, due to their pixel-wise object differentiation, excel at fine-grained distinctions. In contrast, methods like DINO lack such capabilities, suggesting that incorporating segmentation models into our ensemble could be beneficial. We leave this as a direction for future work.
> > Also, as you suggested, we have clarified the networks and training data in Fig. 11. Fig. 11 is included not because we intended to manipulate performance comparisons but because the existing methods fail to work under our settings. Fig. 13 illustrates this clearly: when using the method in [5], which updates using only a discriminative model, generating from noise produces results resembling noise.
> >
> > Q4.  We will make sure to clearly articulate the similarities and differences between our work and dataset distillation in the revised version.
> >
> >
> >
> > [1] Y. Chen et.al, Analyzing Generalization of Neural Networks through Loss Path Kernels (NeurIPS 2023)
> >
> > [2] Ziwei Ji and Matus Telgarsky, Directional convergence and alignment in deep learning (Neurips 2020)
> >
> > [3] Yun, Chulhee, Shankar Krishnan, and Hossein Mobahi. “A Unifying View on Implicit Bias in Training Linear Neural Networks.” International Conference on Learning Representations, 2021, https://openreview.net/forum?id=ZsZM-4iMQkH.
> >
> > [4]Kumari, Nupur, et al. “Ensembling Off-the-Shelf Models for GAN Training.” Proceedings of the IEEE/CVF Conference on Computer Vision and Pattern Recognition (CVPR), 2022, pp. 10651-10662.
> >
> > [5] Image synthesis with a single (robust) classifier.

---

### Official Review · Reviewer_vRMK · 2024-11-05

**Soundness:** 3
**Presentation:** 2
**Contribution:** 2
**Rating:** 5
**Confidence:** 3

**Summary:**

The paper proposes a generative approach that uses pre-trained discriminative models (such as DETR, DINO, ResNet) for image generation. The proposed approach learns to generate samples by calculating the distance between the data distribution and sampled point using a kernel inspired by Neural tangent Kernel and shows similarity to score function used in diffusion models. The paper proposes Discrimination for Generation ( DFG) approach that could be used in conditional and unconditional generation setting. The paper also shows analysis of how the proposed approach is equivalent to approximating score function in diffusion models. The paper shows qualitative results on generative/in-painting tasks to demonstrate the efficacy of the proposed approach.

**Strengths:**

The paper explores an approach that transforms a pre-trained discriminative model (in domains of image segmentation, detection, classification) into generative models. The approach explains how we can use an approximated kernel to compute the distance between sampled data points and data distribution. The proposed approach shows results on different types of discriminative models for conditional as well as unconditional generation. The paper provides theoretical analysis to explain the proposed distance metric. The paper is easy to follow and shows results on in-painting and explanation tasks.

**Weaknesses:**

It would be helpful for a reader to get a better understanding of the following:

1. It would helpful for the reader to see some discussions around the time complexity of each inference call and some comparison with other baselines. It would be great to see specific runtime comparisons and complexity analysis with baselines such as SOTA diffusion models(eg. SDv2) and GANs (eg. StyleGANs)

2. Quantitative evaluation metrics (FID, IS) on standard datasets (e.g. CIFAR-10, ImageNet) with other generative baselines (diffusion models - stablediffusion v2.1 or XL, GANs - styleGANv2) could help the readers in evaluating the efficacy of the approach as the generated sample don’t look like high quality as mentioned in the paper.


3. It would helpful for the reader to see some discussions around Energy based models and even other approaches that leverage discriminative models for image generation. For example, discussion around referred energy-based [1] or discriminative generation methods [2] and how DFG approach compares or differs.

 It would be helpful for the readers if some more related generative papers are added as references:
 [1] Duvenaud, David, et al. "Your classifier is secretly an energy based model and you should treat it like one." ICLR 2020
[2] Santurkar, Shibani, et al. "Image synthesis with a single (robust) classifier." Advances in Neural Information Processing Systems 32 (2019)

Minor typo

line 103 similarly -> similar

**Questions:**

Should there be a negative sign in equation 9 instead of addition sign?

---

> ### Author Response · Authors · 2024-11-21
>
> First of all, I appreciate you for recognizing some of our contributions: utilizing different types of discriminative models as generative models and providing a theoretical justification for using the Neural Tangent Kernel (NTK) as a distance measure. However, as a kindly reminder, there is a distinction—our approach directly uses the NTK, and we use weight as its proxy. I also understand your concerns and am more than willing to address them.
>
> ---
>
> #### **Q1. It would be helpful for the reader to see some discussions around the time complexity of each inference call and some comparison with other baselines. It would be great to see specific runtime comparisons and complexity analysis with baselines such as SOTA diffusion models (e.g., SDv2) and GANs (e.g., StyleGANs).**
>
> Our model takes approximately **1 hour and 20 minutes to generate 200 images on a single A100 GPU.** Since no optimization was applied during implementation, there is room for improvement. Regarding the computation cost, each iteration consumes $O(pd)$, where $p$ is the number of parameters, and $d$ is the dimension of the generated images.
>
> ---
>
> #### **Q2. Quantitative evaluation metrics (FID, IS) on standard datasets (e.g., CIFAR-10, ImageNet) with other generative baselines (diffusion models - Stable Diffusion v2.1 or XL, GANs - StyleGANv2) could help the readers in evaluating the efficacy of the approach as the generated samples don’t look like high quality as mentioned in the paper.**
>
> Thank you for your valuable feedback. While detailed quantitative evaluations are included in our global response, we’d like to highlight the key points here:
>
> - **Our method significantly outperforms similar prior works** that aim to generate images using discriminative models. These prior methods often fail to produce proper image generations, requiring modifications to the classifier and being limited to conditional image generation, as noted in the global response.
> - Unlike these approaches, **our method generates images directly from noise without strict constraints** or modifications to the discriminative model. It achieves substantially higher performance with off-the-shelf models, surpassing the quality of previous methods.
> - While our performance does not yet reach the levels of advanced generative models like StyleGAN2 or Stable Diffusion, it **outperforms early iterations of these models** such as initial VAEs and GANs, which were constrained to low-resolution datasets like CIFAR-100.
> - This highlights the **breakthrough potential of our approach,** which demonstrates that discriminative models can serve as effective generative models, challenging traditional paradigms.
>
> For additional details and quantitative metrics, please refer to our global response.
>
> ---
>
> #### **Q3. It would be helpful for the reader to see some discussions around Energy-Based Models and even other approaches that leverage discriminative models for image generation. For example, discussion around referred energy-based [1] or discriminative generation methods [2] and how DFG approach compares or differs.**
>
> Thank you for pointing out some fields that seem relevant to our work—energy-based models (EBMs) and discriminative generation methods. We appreciate your insight and have provided additional comparisons in the global response. Here, we briefly address the distinctions:
>
> - **Energy-Based Models (EBMs):**
>   EBMs, such as Joint Energy-Based Models (JBMs), are generative models by design. For instance, JBMs explicitly aim to model $P(X, Y)$, and during training, they generate images and assign losses to them, creating a generative process inherently. This approach fundamentally differs from ours, which uses discriminative models directly for generation without building a generative model during training.
>
> - **Discriminative Generation Methods:**
>   These methods typically minimize objectives such as:
>
>   $$\arg\min_x CE(f(x), y) + \text{reg}(x)$$
>
>   However, this does not satisfy the **necessary and sufficient conditions for true generation,** as it does not guarantee that the generated data lies on the manifold $\mathcal{M}_y$. Instead, it only ensures classification consistency, which is vulnerable to adversarial attacks. To address this, these methods often rely on additional training or initialization steps (e.g., multivariate Gaussian sampling for class probabilities), which our method avoids entirely.
>
> In contrast, **DFG generates data directly from noise, adhering to the manifold geometry, without relying on additional initialization or specialized training steps.**
>
> ---
>
> #### **Q4. Should there be a negative sign in equation 9 instead of an addition sign?**
>
> The change from a negative sign to an addition sign is correct, as Equation 9 measures similarity, and we need to ensure they are similar.

---

> > ### Comment · Reviewer_vRMK · 2024-11-23
> > **comment response**
> >
> > Thanks a lot for your detailed response on the quality comparison with other generative baseline models and the discussion about computation cost. It would be great to see this being referred in the manuscript too. Therefore, I have increased my rating to 5.

---

> > > ### Author Response · Authors · 2024-11-25
> > > **Comparison will be included**
> > >
> > > Dear Reviewer vRMK,
> > >
> > > Thank you for your positive feedback.
> > > Are there any other issues that make you hesitate to further increase the score to an acceptance level?
> > > I am asking this since 5 pts is below the acceptance level.
> > >
> > > Sincerely,
> > >
> > > Authors

---

### Author Response · Authors · 2024-11-21
**Table of the Qualitative and Quantitative Comparisons with Baselines**

|                                     | Does not Require data in training | Does not Require Generative models | Does not train a model | Theoretical Guarantee | Can generate in ‘any’ discriminative model | Unconditional Generation | Resides in Manifold |
|-------------------------------------|------------------------------------|-------------------------------------|-------------------------|-----------------------|---------------------------------------------|--------------------------|---------------------|
| Feature Inversion (XAI)            | O                                  | O                                   | O                       | X                     | X                                           | X                        | X                   |
| Energy-Based model                 | X                                  | X                                   | X                       | X                     | X                                           | X                        | O                   |
| Discriminative generative model    | X                                  | O                                   | X                       | X                     | X                                           | X                        | X                   |
| Dataset Distillation               | X                                  | O                                   | X                       | X                     | X                                           | X                        | X                   |
| Generative model                   | X                                  | X                                   | X                       | X                     | X                                           | O                        | O                   |
| DFG (Ours)                         | O                                  | O                                   | O                       | O                     | O                                           | O                        | O                   |
---

Table A1. A Qualitative comparison of our algorithm and comparable counterparts

---
|Paper                                                                                             | Quality Score |
|-------------------------------------------------------------------------------------------------------------|---------------|
| Multi-Target Tracking with Transferable Convolutional Neural Networks                         | 0.00097       |
| Your classifier is secretly an energy based model and you should treat it like one               | 0.00376       |
| Image synthesis with a single (robust) classifier                                       | 0.02398       |
| A Style-Based Generator Architecture for Generative Adversarial Networks                      | 0.4790        |
| Denoising Diffusion Probabilistic Models                                                | 0.3389        |
| Discrimination for Generation: Turning Discriminative Models into Generative Models  (Ours)     | 0.0678        |
---

Table A2. A quality score for our algorithm and comparable counterparts. Quality score is from Q-align[1]. The reason for using Q-align is that it achieves state-of-the-art performance across all quantitative measures, enables calculations with a small amount of data, and provides a unified score in a dataset-agnostic manner.

---

[1] Wu, Haoning, et al. “Q-Align: Teaching LMMs for Visual Scoring via Discrete Text-Defined Levels.” Proceedings of the 41st International Conference on Machine Learning (ICML 2024), 2024, https://arxiv.org/abs/2312.17090.

---

> ### Author Response · Authors · 2024-11-21
> **The field of DFG and contribution Recap**
>
> First and foremost, we would like to express our sincere gratitude to all reviewers for taking the time to thoroughly read our paper and provide valuable feedback. We are especially grateful for recognizing the elegance of our methodology, its potential as a novel generative model, and our contribution as the first to bridge discriminative and generative models. Across the reviews, there were common questions regarding the baselines to compare our work against, quantitative metrics, and theoretical aspects of our approach. To address these points, we will provide a global comment summarizing the common questions and clarifying our contributions in relation to existing research. Additionally, we have updated the supplementary materials to include **Figures 11–15**, which provide comparisons between our approach and baselines, as well as a conceptual diagram comparing our methodology with diffusion models.
>
> ---
> ## We generate without generative model
>
> Our methodology is the **first** to achieve generation in the strict sense using only a discriminative model, outperforming existing methods that rely on additional favorable conditions. Unlike previous approaches that required conditional generation, semi-generative models, or modifications to classifiers to facilitate generation, our method operates without any of these adjustments. We simply use an off-the-shelf discriminative model, such as ResNet, and generate directly from noise, without relying on reference images or any preconditioning. Despite these minimal requirements, our results surpass those of methods that benefit from significant additional setup (See Fig.11). In fact, our method already shows better performance than early generative models. As you can see in the attached supplement, our methodology is **much better** than existing methods at a glance, and in terms of performance, it achieves at least **three times**, and up to **100 times** higher quality scores than previous methods (Table A2 above).
>
>
> ---
> ## Contribution Recap
>
> We have summarized our contributions in **Table A1**. As you can see from this table, we compared DFG to similar counterparts like feature visualization, discriminative generative models, and energy-based models. The most notable point is that **unconditional generation** is possible, because existing methods could only perform conditional generation on a label $y$ . As the goal of a generative model is to model $p(x)$ , existing methods were not true generative models. In the case of energy-based models, they are actually a generative model  because they have a loss function that creates a generative model. Furthermore, **only our methodology has a theoretical guarantee**. That is, our method is based on the assumption of ‘generalization in discriminative model’ which is one of the most widely and extensively researched and well-established. But other methods do not have bounds or theories explaining why generation actually occurs. Except for generative models, **only DFG achieves sampling on the manifold**, meaning it samples data that resembles actual data, which is impossible for other similar methods.
> For feature visualization, which is mentioned by **reviewer fTxU**, this method actually generates images by  gradient ascent. That is, utilizing  $\nabla_x L(f(x), y)$ . However, such methods cannot generate real images. This method starts from noise and iteratively adjusts the noise to increasingly predict label  $y$ , but even if it predicts  $y$, we cannot say that the image created is an image of $y$  because the space where images exist is **extremely sparse**. In fact, this method is developed from adversarial attacks and is a method used to demonstrate black-box features of deep models. Using a method intended to show black-box as an explanation is contradictory. In discriminative generative models, which also uses gradient descent, they tried to solve this problem by making the classifier robust, but even this method executed with the seed initially made close to the image manifold. They actually created a pseudo-generative model by making the dataset into mixed Gaussians and then sampled from there to create images. If we try to make it like other generative models, as shown in **Fig. 13**, no images are generated at all.

---

> ### Author Response · Authors · 2024-11-21
>
> ## DFG Shows That ‘Any’ Discriminative Model Can Be Used to Compute the Score in Diffusion Models
>
> As mentioned in **Section 4.1**, our method is a generalization of score $\nabla\log p(x)$, and diffusion models are actually a model about predicting score. We showed that we can compute that score using only general discriminative models. **We think that we failed to convey this idea to the reviewer, so we want to specify this procedure.**
>
> First, probability $p(x)$ can be seen as a generalization of **Kernel Density Estimation (KDE):**
>
> $$\hat{p}(x) = \frac{1}{n} \sum_{i=1}^n K(x, x_i)$$
>
> which generalizes to:
>
> $$p(x) = \int_M K(x, u) \, du$$
>
> As a result, the **score metric** can be written as follows:
>
> $$\nabla_x\log p(x) = \nabla_x \log\int_M K(x, u) \, du$$
>
> Simply put, in the case of the gradient of the density function, this is a **geometric interpretation** that pulls $x$ toward the manifold where the data actually lies. When approached geometrically, we no longer need to take the logarithm for computational convenience and stability, and our DFG ultimately solves the **generalization** of the following equation:
>
> $$\nabla_x \int_M d(x,u) \, du$$
>
> We provide the **conceptual diagram in Fig. 12.** The notable difference is the usage of a **static update step $\Delta$.** This appears to be the reason for the **high frequency or artifacts** mentioned by **Reviewer fTxU** and **7jxH.** The high frequency refers to the fine details of the images, and the same update step might lead to an overexaggeration of these details.
>
> ---
>
> ## Theoretical Guarantee of DFG
>
> **TL;DR**: Our methodology is based on the **theoretical guarantee** of the **generalization of discriminative models** (which generative models do not possess). Specifically, we generalize the probability measure in score-based models to a **general measure** (see Fig 12) and implement this using discriminative models.
>
> **Reviewer fTxU, 7jxH** questioned theoretical guarantees, and our guarantee directly utilizes the **well-defined generalization bounds of discriminative models.** Considering that generative models lack such theoretical guarantees, this is a significant advantage of DFG. **In other words, DFG has a theoretical guarantee, while generative models do not.**
>
> To elaborate on what we stated in **line 217-223,** the theoretical definition of generalization in discriminative models is as follows: if the error on $(X_{\text{train}}, Y_{\text{train}})$ is $a$, then the error bound on $(X_{\text{test}}, Y_{\text{test}})$ is $ca$ such that $X_{\text{test}} \sim \mathcal{D}$, which implies the following: as long as $X_{\text{train}}$ is well-sampled from the **manifold $\mathcal{M}$** where the data lies, the model will achieve the same bounded performance on $X_{\text{test}}$. Geometrically, this means $X_{\text{test}} \in \mathcal{M},$ where $\mathcal{M}$ is the space in which the data resides. **This is something we inherently accept as true and is one of the most robust assumptions in deep learning.** Our methodology builds directly upon this **theoretical and empirical foundation.**
>
> ---
>
> ## Using a Functor
>
> In simpler terms, the equations can be interpreted as follows: even if we trained on different datasets $A$ and $B$, as long as $A, B \subseteq X$ and are sampled from the **same manifold $X$,** we will obtain the **same model** for both cases. Our ultimate goal can be summarized as:
>
> $$f(\cdot; X_{\text{train}}) \simeq f(\cdot; X_{\text{ours}}).$$
>
> Thus, as mentioned in **Sec 4.4,** we minimize the distance based on this context. In this case, as **Reviewer fTxU, 7jxH** questioned, DFG does not fall into **stationary or edge cases.** These cases usually occur in solving optimization problems, **not in distance minimization.**

---

> > ### Comment · Reviewer_7jxH · 2024-11-24
> >
> > Thank you for the detailed explanation. These points are crucial for understanding the paper and should be included in the final manuscript. However, there is a confusion with this statement "When approached geometrically, we no longer need to take the logarithm for computational convenience and stability, and our DFG ultimately solves the generalization of the following equation:". In Figure 12, there are inconsistent notations between the diffusion model and the proposed method. Additionally, the logarithm is still included in the figure, which conflicts with the statement.

---

> > > ### Author Response · Authors · 2024-11-27
> > >
> > > Thank you for your feedback. As you pointed out, not using the logarithm is correct, and this was an oversight on our part. We have updated the figure and its caption accordingly. I also agree that the geometric aspect should be explained in the main section, and I plan to revise this in a future update. For the inconsistent notation, I have added a detailed explanation in the caption to address this issue. Thank you again for helping improve the paper!

---

### Meta-Review · Area_Chair_jUr3 · 2024-12-21

**Metareview:**

The paper presents a generative approach that utilizes pre-trained discriminative models, including DETR, DINO, and ResNet, for image generation. The proposed method generates samples by computing the distance between the data distribution and sampled points using a kernel inspired by the Neural Tangent Kernel (NTK). The authors demonstrate the effectiveness of their approach through various tasks, including unconditional and conditional generation, inpainting, editing, and data analysis

The AC acknowledges the novelty of exploring NTK alignment for generative tasks in the context of discriminative models. However, the paper falls short in several critical areas. It lacks sufficient evaluation, including comparisons with other generative models on standard datasets using established evaluation metrics. Furthermore, while there has been substantial prior work on utilizing discriminative models for generation, the paper fails to adequately discuss these works or establish meaningful connections to the existing literature.

The AC believes the paper requires significant improvements before it is suitable for publication and recommends rejection in its current form.

**Additional Comments On Reviewer Discussion:**

Reviewer vRMK expressed concerns about the lack of quantitative evaluation metrics, such as FID and IS, on standard datasets, insufficient comparisons with other generative models, and the absence of discussions regarding computational costs. While the authors’ feedback partially addressed these concerns, Reviewer vRMK ultimately marginally rejected the paper.

Reviewer fTxU similarly raised issues, including the absence of quantitative evaluations like FID and human assessments, along with errors in the manuscript. They also highlighted significant methodological similarities with Dataset Distillation works that warrant discussion. However, the authors failed to adequately address these concerns, leading Reviewer fTxU to marginally reject the paper.

Reviewer 7jxH accepted the paper but recommended incorporating quantitative metrics for image generation to enhance its quality. Despite the results being less compelling than those from other generative approaches, Reviewer 7jxH valued the paper for providing novel insights into the representation space linking generative and discriminative models.

The AC emphasized that the lack of quantitative results, such as FID scores, to evaluate the method’s performance is a critical concern shared by all reviewers. Furthermore, the AC noted the need for stronger discussions and connections to prior work. The paper requires substantial revisions before resubmission.

---

### Decision · Program_Chairs · 2025-01-22

Reject